# Bacterial Succinoglycans: Structure, Physical Properties, and Applications

**DOI:** 10.3390/polym14020276

**Published:** 2022-01-11

**Authors:** Jae-pil Jeong, Yohan Kim, Yiluo Hu, Seunho Jung

**Affiliations:** 1Department of Bioscience and Biotechnology, Microbial Carbohydrate Resource Bank (MCRB), Konkuk University, Seoul 05029, Korea; bruce171525@gmail.com (J.-p.J.); shsks1@hanmail.net (Y.K.); lannyhu0806@hotmail.com (Y.H.); 2Department of Systems Biotechnology, Institute for Ubiquitous Information Technology and Applications (UBITA), Center for Biotechnology Research in UBITA (CBRU), Konkuk University, Seoul 05029, Korea

**Keywords:** succinoglycan, bacterial polysaccharides, application, hydrogels, biomaterials

## Abstract

Succinoglycan is a type of bacterial anionic exopolysaccharide produced from *Rhizobium*, *Agrobacterium*, and other soil bacteria. The exact structure of succinoglycan depends in part on the type of bacterial strain, and the final production yield also depends on the medium composition, culture conditions, and genotype of each strain. Various bacterial polysaccharides, such as cellulose, xanthan, gellan, and pullulan, that can be mass-produced for biotechnology are being actively studied. However, in the case of succinoglycan, a bacterial polysaccharide, relatively few reports on production strains or chemical and structural characteristics have been published. Physical properties of succinoglycan, a non-Newtonian and shear thinning fluid, have been reported according to the ratio of substituents (pyruvyl, succinyl, acetyl group), molecular weight (M_w_), and measurement conditions (concentration, temperature, pH, metal ion, etc.). Due to its unique rheological properties, succinoglycan has been mainly used as a thickener and emulsifier in the cosmetic and food industries. However, in recent reports, succinoglycan and its derivatives have been used as functional biomaterials, e.g., in stimuli-responsive drug delivery systems, therapeutics, and cell culture scaffolds. This suggests a new and expanded application of succinoglycan as promising biomaterials in biomedical fields, such as tissue engineering, regenerative medicine, and pharmaceuticals using drug delivery.

## 1. Introduction

Polysaccharides are naturally abundant substances that can be extracted from sources such as plants, animals, microorganisms, etc. [1]. Polysaccharides are widely investigated with the following physical and biological properties, such as high conductivity, cost-effectiveness, biodegradability, biocompatibility, and low immunogenicity. These properties could explain why polysaccharides were used in food, cosmetic, medical application, etc. [2,3]. Moreover, each polysaccharide has its own functional groups (e.g., primary alcohol group, carboxylic group, amine group, etc.) which can be simply modified by an organic reaction. Due to these characteristics, polysaccharides are widely used as a platform in applications such as green electronics, nanoparticles, and hydrogels through chemical modification [3,4,5,6,7,8,9,10].

Currently, most polysaccharides used in industry are obtained from plants and algae because they are easy to process and effective in refining processes. Recently, polysaccharides extracted from microorganisms have emerged as a source of new biomaterials. They have excellent biocompatibility, renewability, biodegradability, easy availability, stable production cost, constant and reproducible physiological properties [11]. Due to the unique physical properties that only bacterial polysaccharides can possess, they have emerged as a biomaterial that has recently been in the spotlight [12,13,14,15]. Bacterial polysaccharides can generally be divided into three groups: intracellular polysaccharides, cell wall polysaccharides, and extracellular polysaccharides. Among them, especially extracellular polysaccharides (EPS) are defined as polysaccharides produced by microorganisms and secreted out of cells. The roles of bacterial EPS are diverse, such as bacterial adaptation to external environmental changes, biofilm formation, and cell signaling. In particular, in the rhizosphere, polysaccharides have been reported to be involved in cell signaling reactions between plants and microorganisms, such as plant root nodulation for nitrogen fixation as well as biofilm formation [16,17,18]. As well as these inherent roles of bacterial EPS, the different structural and functional properties of EPS that can be produced from each bacterial strain have great potential as biomaterials in the fields of food, cosmetics, pharmaceutical, chemical, and biomedical industries [19,20,21,22,23,24,25,26]. It is the strong advantage of bacterial EPS that the same kinds of polysaccharides can be rapidly and efficiently obtained with enhanced yields and controllable compositions by culturing under tunable conditions compared with plant- and algae-extracted polysaccharides [27]. Based on these advantages, studies concerning the production of bacterial polysaccharides through bioreactors for mass production are currently ongoing, and many bacterial EPS, such as cellulose, xanthan, pullulan, gellan, and curdlan, have been used in laboratories and industries [28,29,30,31,32,33,34].

Succinoglycan, one of the bacterial EPS, is an anionic and water-soluble extracellular polysaccharide produced by *Rhizobium*, *Agrobacterium*, and other soil bacteria. This anionic exopolysaccharide plays an important role in building an effective system of symbiotic nodules, especially with legumes [35,36,37,38,39,40,41,42], because it is highly stable in high temperature and pressure and suitable for a wide range of applications in other industries, including foods, pharmaceuticals, and thickeners, stabilizers, emulsifiers, texture treatments, and gelling agents, under optimized operating conditions, like high shear rates and extreme pH [43,44]. Table 1 shows general information regarding the physiochemical properties, and some applications of various anionic bacterial EPS. Unlike other anionic bacterial EPS currently being studied, succinoglycan is not easy to mass-produce and the behavior of structure-based polymers is not well known. However, many physiological studies related to nitrogen fixation and biofilm formation including succinoglycan have been conducted, and recent succinoglycan application studies suggest that it can be effectively used as a platform for future biomaterials [45,46].

Various bacterial strains that produce succinoglycan, chemical analysis methods related to the physical and structural properties of succinoglycan, and recent biotechnological applications of succinoglycan will all be discussed in this review.

## 2. The Structure and Composition Analysis of Succinoglycans Extracted from Bacteria

As a bacterial EPS, succinoglycan is found mainly in the *Rhizobium*, *Agrobacerium*, *Psedomonas* species as one of the biofilm components. Some separation and purification methods are required to extract succinoglycan EPS. A description of the procedures for a separation and purification method is illustrated below in Figure 1.

A summary of chemical structures and the structural analysis method of succinoglycan EPS extracted from bacteria are shown in Table 2.

### 2.1. Sinorhizobium meliloti

The structural analysis of succinoglycan EPS from *Sinorhizobium meliloti* (*S. meliloti*) has been published in many papers [57,58,59]. It has a repeating octasaccharide unit consisting of non-carbohydrate substituents containing pyruvic acid, succinate, and acetate base in a molar ratio of glucose to galactose 7:1, as shown in Figure 2. ^1^H NMR spectra of succinoglycan EPS explained that the singlet resonances at 1.48, 2.16, and 2.8 ppm were assigned to methyl protons of the pyruvyl, acetyl group, and succinyl groups, respectively. The FTIR spectrum of exopolysaccharide was also analyzed. The characteristic spectrum of succinoglycan was explained through band analysis according to the FTIR spectral analysis. The vibrations of –COO group stretching, C=O asymmetrical stretching, C=O stretching of carbonyl ester, and O–H group stretching were all represented by peaks at 1091.51 cm^−1^, 1095.37 cm^−1^, 1379 cm^−1^, 1626 cm^−1^, and 3351 cm^−1^, respectively [69].

### 2.2. Pseudomonas oleovorans

The structure of succinoglycan EPS(GalactoPol) from *Pseudomonas oleovorans* was confirmed by glycosyl compositional analysis and four major constituent sugar residues were identified: galactose, mannose, glucose, and rhamnose. This EPS was composed of galactose, mannose, rhamnose, and glucose as a molar ratio of 1:0.3:0.06:0.04 respectively. Three kinds of acyl groups were identified after acid hydrolysis of EPS: pyruvate (3.35% by weight), succinate (1.04% by weight), and acetate (0.38% by weight). The FTIR spectrum of succinoglycan EPS was compared with other polysaccharides, such as xanthan, guar gum, and alginate, which have already been reported because their components are composed of mannuronic acid, guluronic acid, galactose, and mannose. Results showed 3400 cm^−1^, 2939 cm^−1^, and 990–1200 cm^−1^, common to all polysaccharides, indicating O–H stretching, C–H stretching of the –CH_2_ groups, and C–H stretching of the polysaccharides, respectively. A band near 1660 cm^−1^ in the guar gum spectrum was assigned to the ring stretching of galactose and mannose. This 1660 cm^−1^ band of guar gum was also observed in succinoglycan EPS (1593–1662 cm^−1^). In addition, a small band at 1732 cm^−1^, which was observed in both succinoglycan EPS and xanthan gum, not found in guar gum, was assigned to an acetyl group. The peak at 1416 cm^−1^ observed for all polymers except guar gum indicated –COO symmetric stretching. Identification of the succinoglycan EPS spectrum for acidic groups was consistent with the presence of acyl groups. A strong similarity of acidic group bands between EPS and xanthan spectra was observed, but this similarity of spectrum is strong with sodium alginate and not with the guar gum spectrum. However, the estimated chemical structure of EPS was not reported. Further study would be needed to confirm the structure of EPS by various structural analysis methods.

### 2.3. Enterobacter Strain A47

The estimated chemical structure of succinoglycan EPS (FucoPol) from *Enterobacter strain* A47 was shown in Figure 3. This EPS was hexamer composed of fucose, galactose, glucose, and glucuronic acid (2.0:1.9:0.9:0.5 M ratio), with the main chain composed of α→4)-α-l-Fucp-(1→4)-α-l-Fucp-(1→3)-β-d-Glcp(1→trimer repeating unit. At the C-3 of the first fucose, a α-d-4,6-pyruvyl-Galp-(1→4)-β-d-GlcAp-(1→3)-α-d-Galp(1→trimer branch is present, with a pyruvate group at C-4 and C-6 of the terminal galactose. This succinoglycan EPS also contained 13−14 wt.% pyruvyl, 3−5 wt.% acetyl, and 2−3 wt.% succinyl in its acyl composition. The structural analysis of succinoglycan EPS (FucoPol) from *Enterobacter strain* A47 was performed by similar methods in the case of galactopol. According to this paper, it was confirmed that the composition of succinoglycan EPS was slightly different depending on the cultivation time. The glycosyl composition analysis explained that its composition was mainly a heteropolysaccharide composed of fucose, galactose, and glucose which had neutral charge. The proportion of the sugar monomers in the purified succinoglycan EPS underwent relatively small changes throughout the incubation period. There was a reduction of the glucose content from 46% to 38%, between days 1.0 and 7.0, while the contents on fucose and galactose increased from 21% to 25% and from 27% to 32%, respectively. The acyl group content of the purified succinoglycan EPS also increased during incubation time, reaching a maximum of 11.71 ± 0.83% of polymer mass by day 7.0. The most abundant acyl groups in succinoglycan EPS were acetyl (6.80 ± 0.66%), pyruvyl (3.90 ± 0.15%), and succinyl (1.01 ± 0.12%). There were other unidentified acyl groups in the polymer, but their content was insignificant. The presence of pyruvyl and succinyl conferred anionic properties to succinoglycan EPS. The polysaccharide analyzed using FTIR spectra of several other commercially available polysaccharides was a bacterial fucose-containing EPS composed of fucose, galactose, and galacturonic acid; alginate, which is an algal polysaccharide composed of mannuronic acid and guluronic acid and acetate; and guar gum, a neutral plant polysaccharide composed of galactose and mannose as a control for structural analysis. A broad and strong band around 3400 cm^−1^, common to all polysaccharides, indicates O–H stretching. This band partially overlapped with the C–H stretching peak of the CH_2_ group appearing at 2940 cm^−1^. The peaks between 1200 cm^−1^ and 900 cm^−1^ indicated skeletal C–O and C–C vibration bands of glycosidic bonds and pyranoid rings. The band of 1720 cm^−1^ observed in succinoglycan EPS and fucogel spectra (not found in guar gum and alginate) could be attributed to C=O stretching of the carbonyl in the acyl groups. Strong bands around 1607 cm^−1^ and 1405 cm^−1^ could be attributed to the asymmetric and symmetric stretching of carboxylates, which were also observed in fucogel and EPS but not in guar gum. Similarly, a distinct band observed at 1564 cm^−1^ could be attributed to C=O asymmetric stretching vibrations of succinate. Identification of bands corresponding to acidic groups in fucose-containing EPS spectra was consistent with the presence of acyl groups, which accounted for 22.4 wt.% of the polymer.

### 2.4. Agrobacterium sp. ZCC3656

The succinoglycan EPS (Riclin) from *Agrobacterium* sp. ZCC3656 has been studied in two structures, crude and purified riclin. Crude riclin was a polysaccharide that had not undergone any pretreatment (Figure 4), and purified riclin was a polysaccharide in which succinate was removed by thermal-alkali treatment. In the case of crude riclin, the mass spectroscopic patterns of 2,3,4,6-Me4-Glcp, 2,4,6-Me3-Glcp, 2,3,6-Me3-Glcp, 2,4,6-Me3-Galp, 2,3,4-Me3-Glcp, and 2,3-Me2-Glcp were observed from the results of PMP-HPLC profile. Based on the analysis of the results, it can be confirmed that the succinoglycan EPS structure was composed of molar ratios of these residues of terminal glucose, 3-linked glucose, 4-linked glucose, 3-linked galactose, 6-linked glucose, and 4,6-linked glucose. The molar ratio of the residues was estimated to be 1.2:2.5:2.9:1.0:1.2:1.1. As a result of ^1^H NMR analysis, anomeric protons were shown in the region of 5–4.5 ppm, confirming the existence of the β-configuration. The ^13^C NMR spectrum contained eight signals in the anomeric region (106–90 ppm), indicating octasaccharide repeat units. There was almost no O-acetyl signal detected in the ^1^H and ^13^C NMR spectra. Because of the high viscosity of the riclin solution, it was hydrolyzed to lower the molecular weight. Thus, it could be confirmed that the structure was composed of octapolysaccharides similar to the pattern of succinoglycan. Analysis of purified riclin was also performed by ^1^H and ^13^C NMR.^1^H spectra showed that each of the signals at δ 5.30 and δ 4.66 was identified as α- and β-anomers protons of the reducing galactose, respectively. Each of the signals at δ 1.48 ppm in ^1^H NMR spectrum and signal at δ 27.32 ppm in ^13^C NMR was assigned to methyl protons of pyruvate, respectively. No succinyl group was detected in NMR spectra because EPS was pretreated with alkaline treatment and heated.

### 2.5. Rhizobium radiobacter Strain CAS

To confirm the substituent groups in succinoglycan EPS, the succinoglycan EPS was analyzed by FTIR spectroscopy. As observed in the spectrum, some characteristic bands were assigned. The band at 3000–3700 cm^−1^ was clearly attributed to the hydroxyl group (O–H) stretching vibration of polysaccharides as well as water adsorption. The band observed at 2924 cm^−1^ was attributed to the asymmetric vibration of C–H function of CH_2_ and CH_3_ groups. Moreover, the peak at 1732 cm^−1^ indicated the presence of C=O stretching of carbonyl ester of acetyl group. Absorption bands around 1406 cm^−1^ were attributed to the carboxylate groups (–COO–) from acidic residues like succinyl groups and the broadband located 1618 cm^−1^ corresponded especially to the stretching vibration of carbonyl group (C=O) of the succinyl and acetyl functional group. The signal observed at 1316 cm^−1^ could be specific to the stretching vibration of carboxylate groups. The specific vibration of methyl group (–CH_3_) could be attributed to the signal observed at 1454 cm^−1^. Finally, the peaks obtained at 1164 cm^−1^ and 1077 cm^−1^ could have resulted from the presence of C–O–C asymmetric and symmetric vibration, respectively, considering that these absorption patterns were very similar to those found in typical bacterial EPS reported previously in the literature. The ^1^H NMR spectrum revealed the presence of main characteristic signals as succinyl- and acetyl group by the methyl protons resonances with chemical shifts of 2.3 ppm and 2.1 ppm respectively. This ^1^H spectrum showed the presence of two acetyl groups and one succinyl group per octasaccharide repeating unit. According to this NMR analysis, no pyruvyl group was observed as a substituent in the succinoglycan EPS from *R. radiobacter*. The type and number of *O*-acyl groups (acetyl and/or succinyl and/or pyruvyl) per octasaccharide repeating unit varied depending on the bacterial strain. The estimated chemical structure of EPS is shown in Figure 5.

## 3. Physical Properties of Succinoglycan

### 3.1. Conformational Analysis

The structure of succinoglycans was examined using the X-ray diffraction (XRD) technique. The results of XRD patterns of the succinoglycans produced by wild type strain, *Rhizobium radiobacter* ATCC 19358 (SG-A), and mutant strain (SG-N) were analyzed. The diffraction patterns for SG-A and SG-N showed broad peaks in the range of 2θ = 20–22. The crystallinities were calculated by integration XRD pattern of SG-A and SG-N. The crystallinity of SG-A and SG-N was 21.96% and 16.09%, respectively. On the other hand, the succinoglycan from *Sinorhizobium meliloti* 1021 strain displayed a broad peak at 2θ = 18.7. Because of poor crystallinity, agarose/succinoglycan hydrogels (AG/SG) showed lower and shifted diffraction peaks at angles of 17.5, 17.4, and 18.0 on the 2θ scale compared to pure agarose [71]. In addition, succinoglycan metallohydrogel using trivalent chromium (Cr^3+^) was verified via XRD measurements [72]. When the concentration of Cr^3+^ was increased from 6.6 mM to 52.8 mM, the crystallinity decreased. This is because succinoglycan changed from a crystalline state to an amorphous state, indicating that intermolecular hydrogen bonding between the hydroxyl/carboxyl group and the metal cation of succinoglycan occurred in the metal hydrogen gel. The results indicated that the addition of Cr^3+^ changed the intrinsic crystallinity of succinoglycan and produced another regular molecular arrangement.

The circular dichroism (CD) spectrum of polysaccharides was used as a tool to suggest the indicator of a secondary structure [73]. In the case of native succinoglycan, a characteristic spectrum with a negative band centered at approximately 200 nm was observed. This band corresponded to the n → π* transition by the carboxyl and carboxylates of pyruvate and succinate. Succinoglycan has a backbone with regular side groups formed by four-*O*-linked glycopyranose residues, wherein the backbone contains two consecutive β-(1,6) glycosidic bonds, one of which is connects the side chain to the main chain. This linkage can give flexibility to the lateral arms of succinoglycans. The fixed charge of pyruvate and succinate attached to this side chain enhance it, and the uncharged backbone is reinforced by the entangled residues of glycofuranose and galtopyranose, creating a regular helix structure [74,75]. Due to the spatial demands of charged bulky side chains, succinoglycans are likely single helices with partial lateral aggregation. Coordination of ferric cations (Fe^3+^) with succinoglycan forms a hydrogel. Conformation of Fe^3+^-coordinated succinoglycan (Fe^3+^-SG) hydrogels were investigated by CD spectropolarimetry. As Fe^2+^ was added to succinoglycan, the n → π* negative transition band due to the carboxyl and carboxylate of succinoglycan weakened. This decrease in the intensity of transition may have been due to the Fe^2+^ complexation of succinoglycan by the binding of Fe^2+^ to the carboxyl groups responsible for the transition. Contrarily, the CD spectra of Fe^3+^-SG showed sharp both positive bands at ~195 and 202–208 nm, and sharp negative bands at ~198 nm. The appearance of these new bands could be attributed to the charge transfer interactions between Fe^3+^ cations and the carboxyl group.

The conformation of succinoglycan macromolecules using atomic force microscopy (AFM) was determined and the obtained data were compared to the measurements obtained in solution [76]. Individual chains and dimers were found in succinoglycan precipitated from pure water, whereas only individual chains were found in 0.01 M KCl. At 0.5 M KCl, succinoglycan formed a gel-like structure on the mica surface [77,78]. Analysis of persistence lengths from the AFM images indicated that succinoglycan became more rigid with increasing ionic strength. Flexible chains corresponding to disordered conformations were observed in water, whereas single-helix chains were imaged at 0.01 M KCl.

### 3.2. Thermal Analysis

The thermogram (TGA) of succinoglycan showed two-stage weight loss [79]. The initial weight loss accounts for the loss of absorbed water molecules attached to the carboxyl groups present in the succinoglycan. The first phase in weight loss is due to the high carboxyl group content, which explains the excellent water affinity and water retention capacity of succinoglycan [80,81]. The second phase in polysaccharide degradation involves the degradation of thermally stable structures formed by crosslinking and strong bonds of succinoglycans. The TGA and differential thermogravimetry (DTG) of succinoglycan obtained under a nitrogen atmosphere with heating rate of 10 °C/min shows a mass loss of about 8.05% at 95 °C in Figure 6a. In the second phase, succinoglycan exhibited a mass loss of 60.64% in the range of 246–370 °C. Succinoglycan solution (1–2%) has non-Newtonian shear-thinning fluid behavior under 25–55 °C. With increasing solution concentrations, viscosity and pseudoplasticity proportionally increase while a temperature increase is inversely proportional to viscosity and pseudoplasticity.

A typical differential scanning calorimetry (DSC) thermogram of a succinoglycan solution is representatively described. The endothermic peak of DSC is generally due to the breakdown of hydrogen bonds and loss of hydroxyl groups when the sample is heated. Thus, the appearance of the endothermic peak is due to the disorder of the structure [82]. The melting temperature (T_m_) typically refers to the transition of the material from a crystalline state to an amorphous state, so the transition temperature corresponds to the melting temperature T_m_. As shown in Figure 6b,c, the T_m_ of succinoglycan is observed as 92.17 °C and the viscosity of succinoglycan decreased rapidly at temperatures above 60 °C. During heating, hydrogen bonds break and the double helix changes its structure, causing the fusion of aggregates and disruption of the network [83]. Thermal properties obtained by DSC support the view that succinoglycans are double helix at 25 °C and semi-flexible in their structural arrangement by melting into single strands above 65 °C.

### 3.3. Rheological Properties

Succinoglycan may have different chemical substituent ratios (pyruvyl, succinyl, acetyl groups) and molecular weight (M_w_) depending on the type of strain and the culture conditions for each strain. Rheological properties, such as viscosity and storage/loss modulus, of succinoglycan can be adjusted depending on the measurement conditions (concentration, temperature, pH, etc.). High viscosity in aqueous solution is one of the main characteristics of this water-soluble polymer [84]. It exhibits an order-disorder transition at a characteristic temperature which depends on the ionic strength, counterion, and polymer concentration [85,86,87,88,89,90].

Succinoglycan has been compared to xanthan, a bacterial polysaccharide with a different molecular weight [91,92]. Although their rheological properties show many similarities, the expansion of the ordered conformation of succinoglycan is greater than that of xanthan [93,94]. The main differences in the behavior of the two polymers can be found in the structural transitions and the rheology of solutions at temperatures above T_m_. Succinoglycan exhibits more abrupt and temperature-dependent conformational transitions, is a more randomly coiling molecule than xanthan above the T_m_, and slowly loses stiffness as it passes through the T_m_. In addition, rheological behavior of a succinoglycan was reported in dilute and semi-dilute solutions as a function of the shear rate, temperature, ionic strength, counterion, succinate content, and conformational structure. Viscosity dependence as a function of molecular weight and lacking succinate substituents was compared with that of the native polymer xanthan.

Several studies on the rheological effect on the difference in functional group and molecular weight of succinoglycan isolated from genetically modified strains have been conducted. The rheological effect of the selective removal of acetyl or succinyl substituents on the function of succinoglycan isolated in genetically modified *Rhizobium meliloti* 1021 was studied [95,96,97,98]. Removal of the acetyl substituent led to a decrease in the order-disorder transition temperature, whereas removal of the succinyl group led to an increase [99,100,101]. Consequently, it was found that removal of the succinyl group dramatically improved the pseudoplasticity of aqueous succinoglycan and increased the cooperativity of the order-disorder transition exhibited by polysaccharides [102,103,104,105]. Recently, a high yield of succinoglycan was obtained from the mutant strain *Rhizobium radiobacter* ATCC 19358 by NTG mutagenesis. Succinoglycan from the wild-type strain (SG-A) has two molecular weights of 1.55 × 10^7^ Da and 1.26 × 10^6^ Da depending on the culture conditions. On the other hand, the mutant succinoglycan (SG-N) was a homogeneous polysaccharide and had a molecular weight of 1.01 × 10^7^ Da. Dynamic frequency sweep tests of SG-A and SG-N showed that the G′ and G″ curves crossed over at 65 °C, indicating a thermal-induced order-disordered transition conformation. From the results for the effect of concentration (2.5–15%) and temperature (25–75 °C) on the apparent viscosity of SG-A and SG-N, succinoglycan solution exhibited non-Newtonian and shear thinning behavior.

Afterwards, in order to increase the production yield of succinoglycan, a study was conducted using a sucrose-based carbon source in *Rhizobium radiobactor* (*Agrobacterium tumefaciens*) strain, and the physical properties of the thus-produced succinoglycan were studied. To investigate possible succinoglycan degradation at elevated temperatures, succinoglycan was dissolved in water at a concentration of 4 g/L and heated in the range of 25 to 90 °C for 5 h. The chemical identity and molecular integrity of the polymers were then confirmed by NMR analysis, rheological measurements, and M_W_ measurements [106,107]. Other studies have reported the structural and rheological studies of succinoglycan prepared by fermentation of sucrose or date syrup under various conditions at concentrations (0.5, 1.0, 1.5, and 2.0% *w*/*w*), temperature (5, 25 and 40 °C), and pH (2.5, 4.0, 7.0, and 10.0). The results exhibited that shear thinning (pseudoplasticity) behavior and the viscosity of succinoglycan of date syrup medium was higher than that of sucrose medium at all tested concentrations. Succinoglycan solution (0.1–1.0%, *w*/*v*) also exhibited non-Newtonian and shear thinning behavior at shear rates that ranged from 0.01 s^−1^ to 1000 s^−1^. A weak gel with a concentration of 0.75% was obtained at room temperature (25 °C). The changes in the storage (G′) and loss (G″) modulus during the heating and cooling cycles indicated that succinoglycan can form a thermo-reversible gel. On the other hand, succinoglycan produced from sucrose (EPS-S) showed lower levels of succinylation and acetylation compared to succinoglycan produced from molasses (EPS-M). According to the TGA thermogram, not only was the melting temperature (T_m_) of EPS-M much higher than that of EPS-S, but EPS-M was more thermally stable than EPS-S. In addition, the succinoglycan showed non-Newtonian and shear-thinning behavior, and the viscosity of EPS-M was higher than that of EPS-S. Production of the succinoglycan using the sugar cane molasses, sucrose, glucose, and lactose as carbon source was analyzed. The molecular weights of succinoglycan from lactose and sugar cane molasses were 2.734 × 10^6^ g/mol and 2.326 × 10^6^ g/mol, respectively. Rheological analysis of the succinoglycan at concentration (0.5–2.0%), temperature (5–75 °C), and pH (2.5–10.0) revealed non-Newtonian and shear thinning behavior.

Succinoglycan has been studied with the aim of developing a mucoadhesive agent. The investigation of the mucoadhesive properties of a series of polymers and their association was carried out by a rheological synergistic approach. The combination of succinoglycan or xanthan with guar hydroxypropyl trimonium chloride and lambda carrageenan is characterized by the highest mucosal adhesion. The mucosal adhesion of succinoglycan was adequate through various approaches, such as rheological synergy, tensile, and washout tests.

In addition, the effect of different extraction methods on production, rheological, and structural properties of succinoglycan were investigated [108]. Eleven different chemical and physical methods were tested for the extraction of succinoglycans from *Rhizobium radiobacter* CAS. Comparing the succinoglycan yields of all methods, the acetone method (3014 mg/L) proved the highest, followed by the cetyl-trimethyl-ammonium-bromide (CTAB 2939 mg/L) and vacuum evaporation (2804 mg/L) methods. Comparing the rheological properties of succinoglycan, succinoglycan recovered by acetone and CTAB method with a shear rate of 50 s^−1^ showed a tendency to make the solution highly viscous with viscosities of 150 and 146 mPa·s, respectively. The results showed that the physicochemical method for EPS extraction had a significant effect on the physical properties.

Recently, it was investigated whether hydrogels could be formed by mixing aqueous solutions of various metal ions, such as K^+^, Na^+^, Ca^2+^, Mg^2+^, Cu^2+^, Zn^2+^, Al^3+^, Fe^3+^, and Cr^3+^, with an aqueous succinoglycan solution. As shown in Figure 7a,b, physical gel formation occurred only when Cr^3+^ and Fe^3+^ solutions were added to the aqueous succinoglycan solution, and this was confirmed by measuring the difference between the storage modulus (G′) and the loss modulus (G″) through a rheological experiment. In addition, Figure 7c shows the results of the reverse vial test for each mixed solution corresponding to a different metal ion. Except for Fe^3+^ and Cr^3^, most metal ions did not induce physical gels. These results suggest that an appropriate amount of trivalent metal ion can coordinate effectively with the carboxyl group of succinoglycan, resulting in a denser and harder hydrogel network.

## 4. Applications of Succinoglycan

### 4.1. Thickener

A thickener is a substance which can increase the viscosity of a liquid without substantially changing its other properties. A recent report on the relationship between succinoglycan and viscosity showed that viscoelasticity increased with increasing succinoglycan EPS concentration. At a concentration of 2% or more, a weak gel is formed, and structural fluidity appears. In addition, an aqueous dispersion of EPS exhibited pseudoplastic behavior. It was found to be stable over time with an increase in shear rate. In addition, EPS was not affected by ions and showed a stable state with no change in viscosity.

### 4.2. Emulsifier

The succinoglycan EPS from *Pseudomonas strain* showed high flocculating capacity (82.6%) among xanthan, guar gum, carboxymethyl cellulose (CMC), and alginate. It also showed that *n*-hexadecane had significant emulsifying activity compared to other carbohydrates. The emulsification index of the *Pseudomonas oleovorans* was investigated against several hydrophobic compounds. Reportedly, succinoglycan EPS was able to stabilize water emulsions with several hydrophobic compounds, including hydrocarbons, vegetable, and mineral oils. It also maintained emulsifying activity during exposure to wide temperature and pH changes as well as to the presence of 2.0 M NaCl. This EPS was also found to have the highest relative stabilization ability among the four polysaccharides presented in the paper. Investigation of the effect of ionic strength on the creaming and agglomeration of polysaccharides confirmed the general destabilizing effect of polysaccharides on low concentrations of food emulsions [109].

### 4.3. Stabilizing Agents

Several studies using succinoglycan to stabilize nanoparticles have been reported. Succinoglycan from *S. meliloti* was used to reduce the silver and stabilize the AgNPs. The metal reduction process can be induced by oxidization of the aldehyde group of reducing sugar of succinoglycan into the COOH group by nucleophilic insertion of hydroxide ion, which further reduces Ag^+^ to Ag (AgNPs). The abundant hydroxyl groups in the succinoglycan can also stimulate the formation of the complex matrix around the AgNPs to stabilize them even without the addition of any reducing agent [110]. Another case of succinoglycan isolated from *S. meliloti* capable of binding silver nanoparticles was reported. The paper mentioned the possibilities of succinoglycan as a stabilizer and for the induction of chemical reactions in the polysaccharide matrix [111] where the function of fucoPol-coated magnetic particles to prevent high stabilization and non-specific binding was studied. The results suggested that fucoPol coating allowed both electrostatic and hydrophobic interactions with the antibody, contributing to enhancing the specificity for the targeted products [112]. In addition, the application of MRI nanoprobes using fucoPol-coated nanoparticles was also performed. It was confirmed that the nanoparticles could be internalized by endocytosis into human cell line HCT116 and ReNcell VM, and these results showed no cytotoxicity to these nanoparticles [113]. Succinoglycan also showed a possibility to be used as a cryoprotective agent. The cryoprotective effect of fucoPol, extracted from *Enterobacter strain* A47, was investigated. The efficacy of succinoglycan as a cryoprotectant was proved in media of different compositions, including those generally used in cell culture and three commercial hypothermic methods. Based on the results, fucoPol was suggested as a promising cryoprotective agent in the field of cell monolayer freezing to organ systems [114]. FucoPol exhibited high photoprotective properties which could be applied to cosmetic industries. FucoPol was photostable and protected epithelial cells and keratinocytes when exposed to UVA, UVB, and visible regions of spectrum. It showed optimal performance at 0.25% regarding UVR neutralization. FucoPol also showed no cytotoxicity and a higher SPF to-concentration ratio than common cosmetic biopolymers [115].

### 4.4. Hydrogel

Recently, hydrogel studies using succinoglycan applied to other polysaccharides have been conducted. First of all, since succinoglycan has a carboxyl group capable of binding to a metal ion, the gelation of succinoglycan can be induced through coordination with the metal ion. For example, research concerning the coordination between succinoglycan and trivalent chromium was reported. The hydrogel was simply prepared by mixing succinoglycan and trivalent chromium. Analysis of TGA and DSC measurements proved that the change in the thermal stability occurred after gelation. Furthermore, if the changes happened after gelation, it had no effects on the preformed hydrogel. This represented that it preserved stabilities when external pH changes occurred. This indicates that stability can be maintained even when external pH changes occur. Another example is the formation of strong gels using the coordination bonding formed between succinoglycans and ferric ions. In this study, the mechanism of their interaction was presented through CD and ATR-FTIR spectral analysis. In addition, it is confirmed that Ferric ion-succinoglycan hydrogel could be changed from gel to sol when external stimuli occurred which could affect the state (oxidation number). FucoPol, a fucose-containing polysaccharide, also had abilities to form a hydrogel bead with cation ion by cation-mediated gelation. The characteristic anions of fucoPol, such as carboxyl, succinyl, and pyruvyl groups, made it possible to induce gelation with ferric and copper cations. The copper-mediated bead had disadvantages, being fragile and cytotoxic. When the copper-containing fucoPol bead was coated with ferric ion-gel layer, it showed a more stable form and had reduced cytotoxicity. The difference of beads was examined according to their morphology, physical properties, and cytotoxicity [116]. As a study for use as a drug delivery, the succinoglycan hydrogel was used as a drug delivery platform via a double cross-linking interpenetrated network (IPN) hydrogel. This hydrogel was synthesized by polymerization between succinoglycan and CMC due to coordination with ferric ion. Its properties such as rheology, cross-section, and pore size could be controlled by the molar ratio of succinoglycan. It showed a significant increase of drug release pattern at physiological pH (pH 7.4) compared to acidic conditions (pH 1.2). This pattern suggested that this hydrogel was suitable for drug carriers under neutral conditions and could be used in the field of controlled drug delivery systems. Succinoglycan was complexed with agarose which had a thermo-responsive property. To induce gelation, succinoglycan and agarose mixture was heated up to 70 °C and cooled to 4 °C. Succinoglycan/Agarose hydrogel constructed a flexible, stable network and showed a good swelling pattern in basic solution. To introduce aldehyde groups to succinoglycan (SGDA), periodate oxidation of succinoglycan was performed. The hydrogel was synthesized via Schiff-base reaction with aldehyde groups of succinoglycan and amine groups of gelatin formed imine bonds. The resulting hydrogel exhibited enhanced mechanical properties and thermal stability. SGDA was also applied to another hydrogel system (Figure 8). Amine group modification of alginate was performed on the carboxyl group of alginate with hydrazine through NHS/EDC coupling. With this hydrazine functionalized alginate, SGDA was crosslinked with Schiff-base reaction. This hydrogel showed self-recoverable, tunable rheological properties, and a pH-responsive degradation pattern. The results indicated the potential of a pH-controlled drug delivery system using succinoglycan.

### 4.5. Biomedical Applications

Nowadays, investigations on functional foods for the pharmaceutical purposes have been conducted using succinoglycan-based polymers themselves. Due to high content of negatively charged group, FucoPol has been reported to have the effect of reducing oxidation caused by Fe^3+^ and successfully neutralizing oxidation caused by H_2_O_2_. It maintained the morphology of Vero cells rapidly exposed to H_2_O_2_, attenuated the decline in viability, and subsequently enhanced their proliferative capacity, as shown in Figure 9 [117]. Riclin dose-dependently inhibited TNF-α, IL-1β, and IL-6 expression in LPS-stimulated RAW 264.7 macrophages. In addition, riclin pretreatment increases the survival rate of D-Gal/LPS treated mice. The results showed that it inhibits serum ALT and AST activity and reduces the production of inflammatory mediators TNF-α, IL-1β, and IL-6. Other study reported that riclin promoted the elimination of *Listeria* in both in vitro and in vivo infection models. The expression and secretion of inflammatory cytokines, including IL-6 and IL-1β, were significantly increased after riclin treatment upon infection. The protective effect of riclin was primarily through MAPK/IL-6 axis activation. Consequently, riclin induced the change of mechanisms during *Listeria* infection, including pathways related to glycolysis, protein synthesis, and oxidative stress. These results suggested that the succinoglycan riclin could potentially be used as a therapeutic antibacterial agent. Riclin reduced the levels of IFN-γ and IL-1β, protecting against streptozotocin (STZ)-caused MIN6 cell injury. Riclin specifically bound to the membrane of macrophages and regulates the ratio of IL-10 and IL-12. Riclin treatment significantly decreased the incidence of STZ-induced hyperglycemia and prevented autoimmune diabetes in non-obese diabetic (NOD) mice. Riclin also showed antitumor activity in sarcoma 180 tumor-bearing mice. Riclin suppressed the tumor growth significantly similar to cyclophosphamide (CTX). Riclin reduced splenocyte apoptosis, as shown by changes in β-cell lymphoma-2 family protein and cleaved caspase-3 protein. This was examined by ^1^H nuclear magnetic resonance (NMR)-based metabolomics analysis and the result revealed that riclin partially altered the metabolic profiles of splenocytes.

## 5. Conclusions and Future Perspectives

Scale-up studies on various polysaccharides are in progress based on the identification of the biosynthetic process of microbial EPS and optimization of culture conditions, and there are successful commercialization cases through large-scale production. Currently, bacterial EPS that can be mass-produced include xanthan gum, dextran, and pullulan, which have an important global market.

Succinoglycan EPS has the advantage that it can have various functional groups depending on the bacterial strains and culture conditions. In addition, there are few studies on what kind of mechanical, chemical, and physical properties will be obtained in response to each functional group attached to the main glycan chain, and on new succinoglycan EPSs derivatives, suggesting that succinoglycan EPS may have potential as a functional biomaterial.

Owing to such advantages and possibility, many attempts have been made to mass-produce succinoglycan by increasing the production yield. The application of succinoglycan, which was used limitedly in the cosmetic and food industries due to its characteristic rheological properties, is now expanding to biomedical fields, such as tissue engineering, regenerative medicine, and pharmaceuticals in the near future.

## Figures and Tables

**Figure 1 polymers-14-00276-f001:**
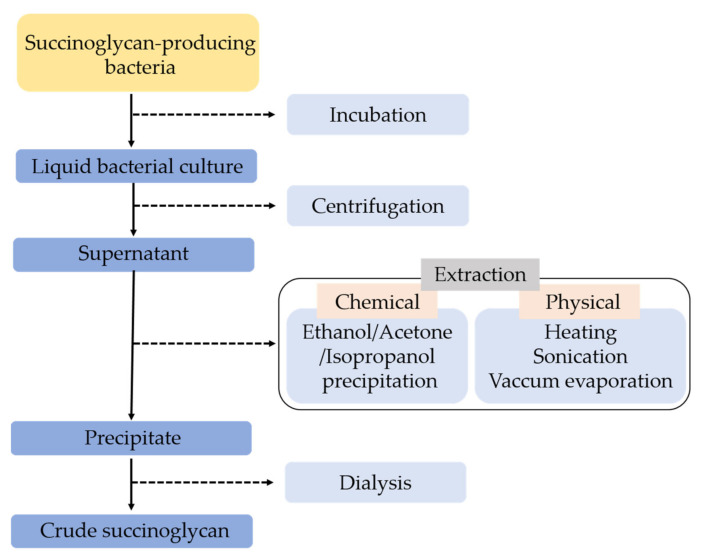
Description of general procedure to extract succinoglycan EPS.

**Figure 2 polymers-14-00276-f002:**
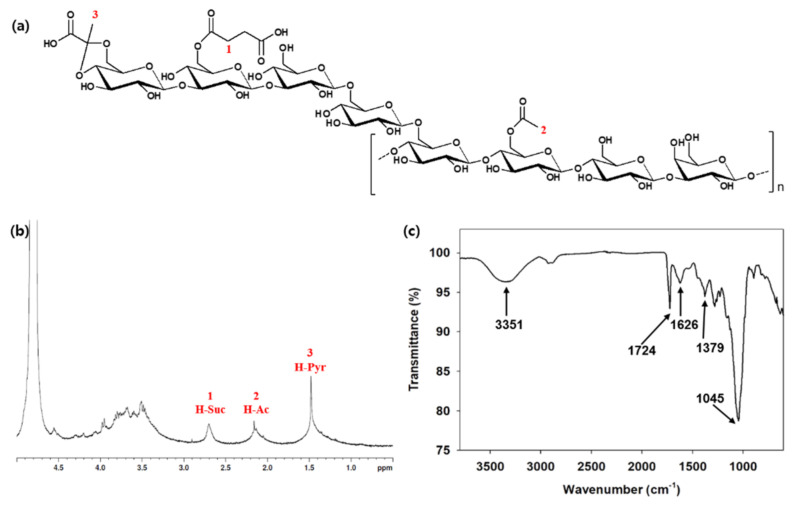
(**a**) Structure of succinoglycan from *S. meliloti*, (**b**) ^1^H NMR spectra of succinoglycan EPS (**c**) FTIR spectra of succinoglycan EPS. Modified from [69,70]. Copyright © 2022 by the authors. Licensed MDPI, Basel, Switzerland.

**Figure 3 polymers-14-00276-f003:**
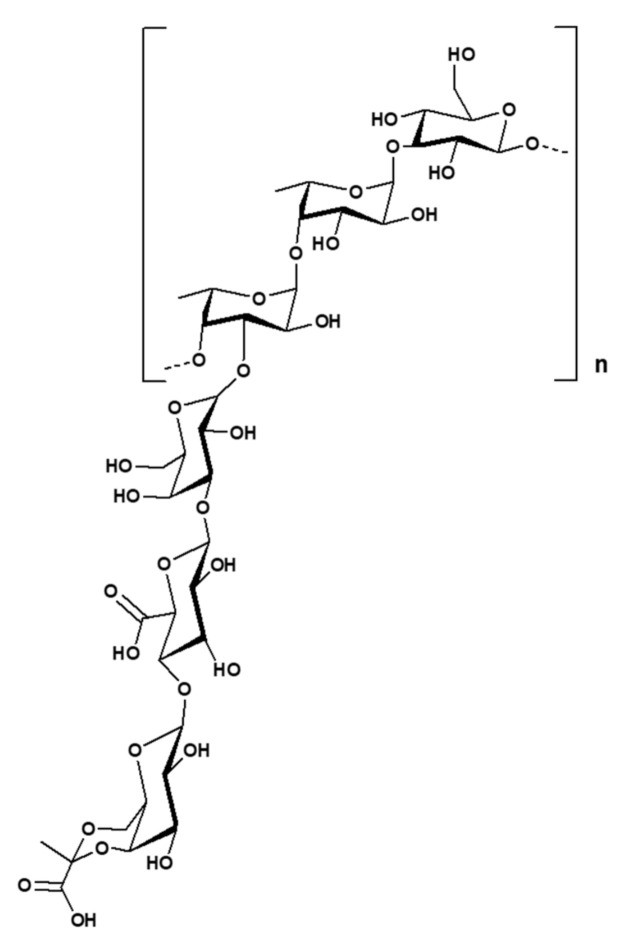
Tentative structure of succinoglycan from *Enterobacter strain* A47. The structure presented here is a deacetylated, desuccinylated form of the biopolymer. Copyright © 2022 by the authors. Licensed MDPI, Basel, Switzerland © 2022.

**Figure 4 polymers-14-00276-f004:**
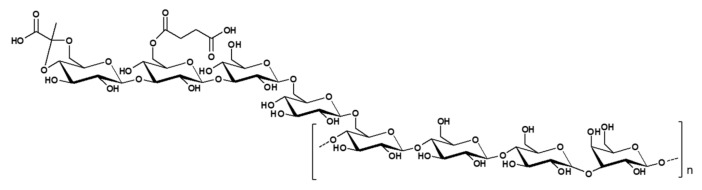
Structure of succinoglycan from *Agrobacterium* sp. ZCC3656.

**Figure 5 polymers-14-00276-f005:**
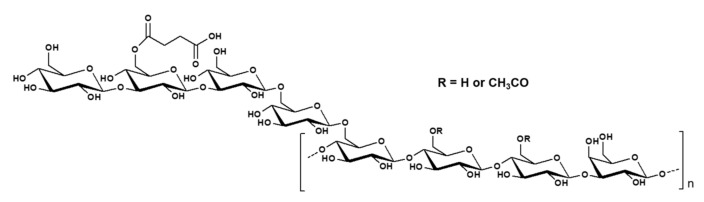
Estimated chemical structure of succinoglycan from *Rhizobium radiobacter* strain CAS resulted from NMR, FTIR analysis.

**Figure 6 polymers-14-00276-f006:**
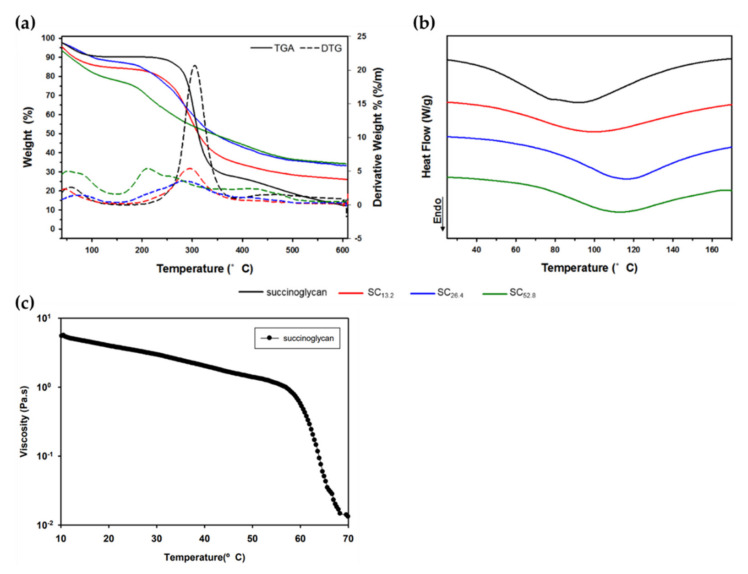
Comparison of the thermal analysis for succinoglycan with different Cr^3+^ concentrations: (**a**) thermogravimetric analysis (TGA) and (**b**) differential scanning calorimetry (DSC). (**c**) Viscosity change during the first heating cycle of 1 wt.% solution of succinoglycan. Copyright © 2022 by the authors. Licensed MDPI, Basel, Switzerland.

**Figure 7 polymers-14-00276-f007:**
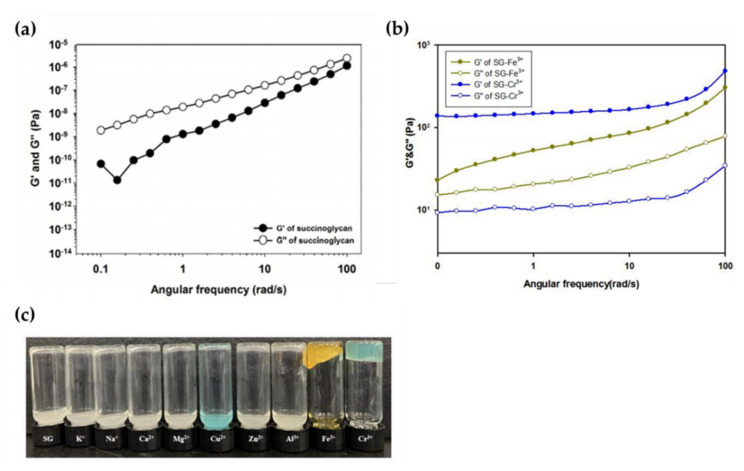
(**a**) Frequency sweep test of succinoglycan (1 wt.%), (**b**) Frequency sweep of aqueous succinoglycan solution containing metal ions Fe^3+^ and Cr^3+^, and (**c**) Gelling performance of succinoglycan with respect to the addition of various metal ions: Inverted vial test after mixing various metal ion solutions (0.25 M) with aqueous succinoglycan solution (2 wt.%). Copyright © 2022 by the authors. Licensed MDPI, Basel, Switzerland.

**Figure 8 polymers-14-00276-f008:**
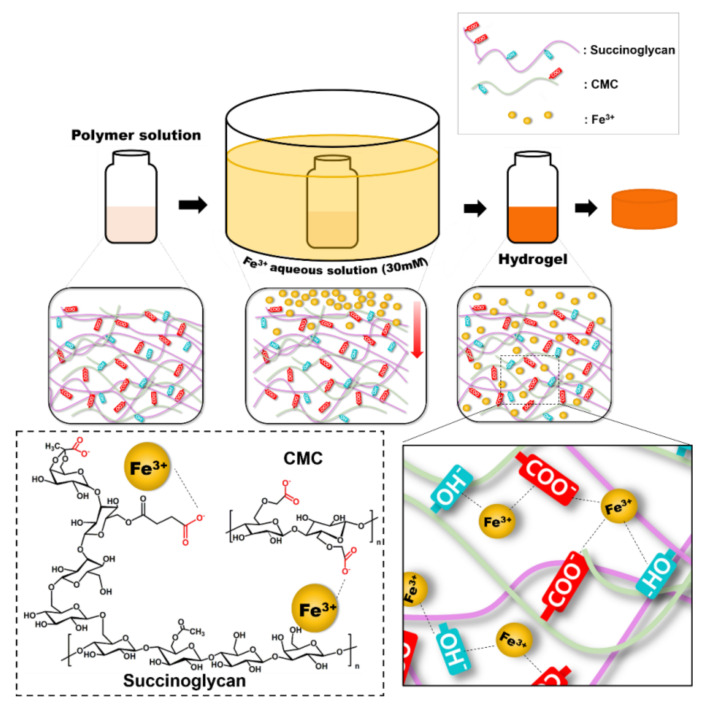
Schematic illustration of model of succinoglycan/carboxylmethyl cellulose interpenetrated network(IPN) hydrogels formation mechanism: Copyright © 2022 by the authors. Licensed MDPI, Basel, Switzerland.

**Figure 9 polymers-14-00276-f009:**
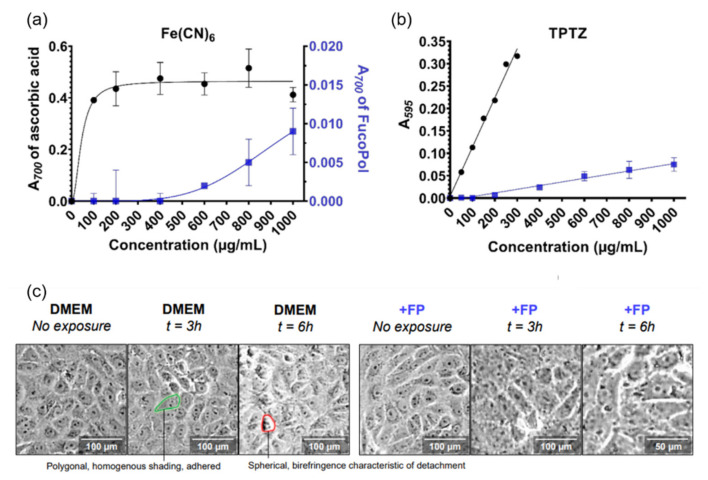
(**a**) FRAP assay using ferricyanide as oxidant species and ascorbic acid (vitamin C) as positive control (black circles). (**b**) FRAP assay using TPTZ as the oxidant species and Trolox (water-soluble vitamin E analog) as the positive control (black). (**c**) Morphologies in the absence of an antioxidant (DMEM) and in the presence of 0.25% FucoPol (+FP) were collected with a 100× magnification. Copyright © 2022 by the authors. Licensed MDPI, Basel, Switzerland.

**Table 1 polymers-14-00276-t001:** Illustration of physiochemical properties and their applications of anionic bacterial EPS.

EPS	Strain	Structure	Components	Molecular Weight (g/mol)	Main Properties	Main Applications	Market Value (US$)	Ref.
							Price (US$)/kg	
Xanthan	*Xanthomonas* sp.	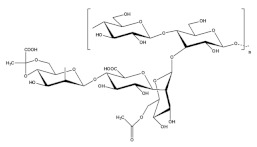	Glucose Mannose Glucuronic acid Acetate Pyruvate	<50 × 10^6^	Hydrocolloid - High viscosity yield at low shear rates and low concentration - Stability over wide temperature, pH and salt concentrations ranges	Foods Petroleum industry Pharmaceuticals Cosmetics Agriculture	987.7 million (2020)	[47,48,49,50]
							3–5	
Gellan	*Sphingomonas paucimobilis* ATCC 31461	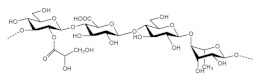	Glucose Rhamnose Glucuronic acid Acetate Glycerate	5.0 × 10^5^	Hydrocolloid Gelling capacity - Stability over wide pH range Thermo-reversible gels	Foods Pet food Pharmaceuticals Agar substitute Gel electrophoresis	15 million	[51,52]
							55–66	
Alginate	*Pseudomonas* sp. *Azotobacter* sp.	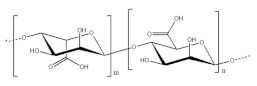	Guluronic acid Mannuronic acid Acetate	<1.3 × 10^6^	Hydrocolloid Film-forming Gellling capacity	Food hydrocolloid Medicine - Wound dressings - Drug release	923.8 million (2025)	[53,54]
							5–20	
Hyaluronan	*Diplococcus* sp. *Streptococcus* sp. *Staphylococcus* sp.	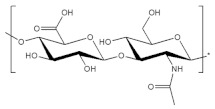	Glucuronic acid Acetylglucosmine	2.0 × 10^6^	Highly hydrophilic Biological activity Biocompatible	Medicine Culture media	15.4 billion (2025)	[55]
							100,000	
Succinoglycan	*Agrobacterium* sp. *Rhizobium* sp. *Pseudomonas* sp.	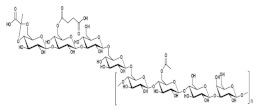	Glucose Galactose Acetate Pyruvate Succinate	LMW < 5 × 10^3^ HMW > 1 × 10^6^	Viscous shear thinning aqueous solutions Acid stability	Cosmetics Foods - Thickener - Emulsifier - Stabilizer Oil recovery	N.A.	[56]

**Table 2 polymers-14-00276-t002:** Chemical structures and analysis methods of succinoglycan EPS.

Strain	Monosaccharide Composition	Acyl Group Composition	Analysis Method	Ref.
*Sinorhizobium meliloti*	Glucose (87.5%) Galactose (12.5%)	Pyruvyl Acetyl Succinyl	Nuclear magnetic resonance spectroscopy (NMR) Electrospray mass spectrometry (ES-MS) Fourier-transform infrared spectroscopy (FTIR)	[57,58,59]
*Pseudomonas oleovorans*	Galactose (70%) Mannose (23%) Glucose (4%) Rhamnose (3%)	Pyruvyl Acetyl Succinyl	FTIR Solid-state NMR Gas chromatography/mass spectrometry (GC/MS)	[60,61,62]
*Enterobacter strain* A47	Fucose (37.8%) Galactose (35.8%) Glucose (17%) Glucuronic acid (9.4%)	Pyruvyl Acetyl Succinyl	FTIR High Performance Liquid Chromatography (HPLC)	[63,64]
*Agrobacterium* sp. ZCC3656	Glucose (87.5%) Galactose (12.5%)	Pyruvyl Succinyl	NMR GC/MS FTIR	[65,66,67]
*Rhizobium radiobacter* strain CAS	Glucose (87.5%) Galactose (12.5%)	Acetyl Succinyl	NMR FTIR High Pressure Anion Exchange Chromatoraphy (HPAEC)	[68]

## Data Availability

Not applicable.

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
