# Peer review of "Bacterial Succinoglycans: Structure, Physical Properties, and Applications"

_polymers, 2022, doi:10.3390/polym14020276_

Round 1

Reviewer 1 Report

Ms. Ref. No. polymers-1510932-peer-review-v2

Title:  Bacterial Succinoglucans: Structure, Physical Properties, and Applications

Polymers

The authors presented an interesting review on the topic of “Bacterial Succinoglucans: Structure, Physical Properties, and Applications”. I think that the review is exhaustive and well-written. However, the review article is suitable for publication in Polymers, however, some issues must be solved before its recommendation for publication, such as:

  • The review addresses a current and fascinating topic, which was clearly and concisely presented. The text is well organized and quite well written, but some major check of English language and grammar needs to be performed.
  •  
  • Following sentences are confusing and should ne repharase
    1. Please replace “It is also known to be suitable for a wide range of applications in other industries such as foods, pharmaceuticals and thickeners, stabilizers, emulsifiers, texture treatments, and gelling agents because it is highly stable in high temperature and pressure. even under operating conditions such as extreme pH and high shear rates.” Line 67-71

with

“Because it is highly stable in high temperature and pressure and suitable for a wide range of applications in other industries including foods, pharmaceuticals, and thickeners, stabilisers, emulsifiers, texture treatments, and gelling agents under optimized operating conditions like high shear rates and extreme pH.”

  1. “In this review, various bacterial strains that produce succinoglycan, chemical analysis methods related to physical, and structural properties of succinoglycan, and recent biotechnological applications of succinoglycan will be introduced.” Line 78-80

with

“Various bacterial strains that produce succinoglycan, chemical analysis methods related to physical and structural properties of succinoglycan, and recent biotechnological applications of succinoglycan will all be discussed in this review.”

  1. “Each band of 1091.51 cm−1, 1095.37 cm−1, 1379 cm−1, 1626 cm−1, and 3351 cm−1 corresponded to the vibrations of -COO group stretching, C=O asymmetrical stretching, the C=O stretching of carbonyl ester, and O-H group stretching, respectively. Line 78-80

With

“The vibrations of -COO group stretching, C=O asymmetrical stretching, C=O stretching of carbonyl ester, and O-H group stretching were all represented by peaks a1091.51 cm−1, 1095.37 cm−1, 1379 cm−1, 1626 cm−1, and 3351  cm−1, respectively.”

  • My major concern is about figures. Overall, they appear to be of very low quality both regarding resolution and conceptualization. In particular, the high resolution of structural images in Table. 1are required.
  • Please add some more figures for a better understanding from different published articles that will be helpful readers.
  • Please do add conclusion and future perspective section in the review article.
  • Harldy30-40% reference has been cited from 2020/2021 literature. It is necessary to cite literature from the last five years (2017-2021).
  • Extensive editing of the English language (grammar, sentence making, spelling, typo errors, etc.) should be made.

Reviewer 2 Report

This manuscript reports the different structures, strains, physical properties and applications of succinoglycans derived from respective bacterial strains. The authors presented structure and composition analysis methods for each of the strains, physical properties and applications and discussed them well. This review would be of greater interest and knowledge for the researchers in the respective field of study. I would recommend the acceptance of this manuscript after minor revision as mentioned below:

  1. Title - the title says 'succinoglucan' and other parts of review say 'succinoglycan' - be consistent in words/spellings throughout manuscript
  2. English check is to be done

Author Response

This manuscript reports the different structures, strains, physical properties and applications of succinoglycans derived from respective bacterial strains. The authors presented structure and composition analysis methods for each of the strains, physical properties and applications and discussed them well. This review would be of greater interest and knowledge for the researchers in the respective field of study. I would recommend the acceptance of this manuscript after minor revision as mentioned below:

☞ Thank you very much for your comments on the recommendation of the acceptance of the manuscript.

1) Title - the title says 'succinoglucan' and other parts of review say 'succinoglycan' - be consistent in words/spellings throughout manuscript

☞ Thank you very much for your comments. As your suggestion, we corrected the word of the title and made the ‘succinoglycan’ be consistent throughout the revised manuscript.

2) English check is to be done

☞ Thank you very much for your comments. As your suggestion, we made extensive editing of the English language.

Reviewer 3 Report

This review provides recent studies about succinoglucan and seem to be valuable for related researchers. Such natural polymers are useful to biomedical application, as authors said in abstract part. To attract many attentions, the figures or illustrations that related to the biomedical applications should be inserted. In this version, only one illustration of IPN hydrogel formation was presented.  It will be so helpful for readers. 

Round 2

Reviewer 1 Report

Ms. Ref. No. polymers-1510932-peer-review-v2

Title:  Bacterial Succinoglucans: Structure, Physical Properties, and Applications

Polymers

The authors presented an interesting review on the topic of “Bacterial Succinoglucans: Structure, Physical Properties, and Applications”. I think that the review is exhaustive and well-written. However, the review article is suitable for publication in Polymers, however, some issues must be solved before its recommendation for publication, such as:

  • My major concern is about figures. Overall, they appear to be of very low quality both regarding resolution and conceptualization. In particular, the high resolution of structural images in Table. 1are required.
  • Harldy30-40% reference has been cited from 2020/2021 literature. It is necessary to cite literature from the last five years (2017-2021).
  • Extensive editing of the English language (grammar, sentence making, spelling, typo errors, etc.) is required that has not been addressed.
  • Please remove references from the “Conclusion and Future perspectives”.
